

**Summer surface air temperature proxies point to near sea-ice-free conditions in the Arctic at 127 ka.**

Louise C. Sime[1], Rahul Sivankutty[1], Irene Vallet-Malmierca[1], Agatha M. de Boer[2], and Marie Sicard[2]

[1]British Antarctic Survey, Cambridge, UK

[2]Department of Geological Sciences, Stockholm University, Sweden.

Correspondence: Louise C. Sime (lsim@bas.ac.uk)





**Abstract.**
The Last Interglacial (LIG) period, which had higher summer solar insolation than today, has been
suggested as the last time that Arctic summers were ice-free. However, the latest suite of Coupled
Modelling Intercomparison Project 6 Paleoclimate (CMIP6-PMIP4) simulations of the LIG produce a
wide range of Arctic summer minimum sea ice area (SIA) results, ranging from a 30% to 96%
reduction from the pre-industrial (PI). Sea ice proxies are also currently neither abundant nor
consistent enough to determine the most realistic state. Here we estimate LIG minimum SIA
indirectly through the use of 21 proxy records for LIG Summer Surface Air Temperature (SSAT) and
11 CMIP6-PMIP4 models for the LIG. We use two approaches. First, we use two tests to determine
how skilful models are at simulating observed proxies for $\Delta$SSAT (where $\Delta$ refers to LIG-PI). This
identifies a positive correlation between model skill and the magnitude of $\Delta$SIA: the most reliable
models simulate a larger sea ice reduction. Averaging the most skilful two models yields an average
SIA of 1.3 mill. $km^2$ for the LIG. This equates to a 4.5 mill. $km^2$, or a 79%, SIA reduction from the PI
to the LIG. Second, across the 11 models, the averaged $\Delta$SSAT at the 21 proxy locations is inversely
correlated with $\Delta$SIA (r = -0.86). In other words, the models show that a larger Arctic warming is
associated with a greater sea ice reduction. Using the proxy record-averaged $\Delta$SSAT of 4.5 ± 1.7 K
and the relationship between $\Delta$SSAT and $\Delta$SIA, suggests an estimated $\Delta$SIA of 4.4 mill. $km^2$ or 77%
less than the PI. The mean proxy-location $\Delta$SSAT is well-correlated with the Arctic-wide $\Delta$SSAT
north of 60°N (r=0.97) and this relationship is used to show that the mean proxy record $\Delta$SSAT is
equivalent to an Arctic-wide warming of 3.7±0.1 K at the LIG compared to the PI. Applying this
Arctic-wide $\Delta$SSAT and its modelled relationship to $\Delta$SIA, results in a similar estimate of LIG sea ice
reduction of 4.5 mill. $km^2$. The LIG climatological minimum SIA of 1.3 mill. $km^2$ is close to the
definition of a summer ice-free Arctic, which is a maximum sea ice extent less than 1 mill. $km^2$. The
results of this study thus suggest that the Arctic likely experienced a mixture of ice-free and near ice-
free summers during the LIG.



## 1. Introduction

The rapid decline in Arctic sea ice over the last 40 years is an icon of contemporary climate change. Climate models have struggled to fully capture this sea ice loss (Notz and Community, 2020), which can sometimes reduce confidence in their future projections (*e.g.* IPCC, 2021). One line of investigation to address this problem, that has not been fully exploited, is the use of past climates to provide information on the future (e.g. Bracegirdle et al., 2019). Investigating the physics and causes of sea ice change, concentrating on Arctic changes during the most recent warm climate periods can help us address this problem (Guarino et al., 2020b). Interglacials are periods of globally higher temperatures which occur between cold glacial periods (Sime et al., 2009; Otto-Bliesner et al., 2013; Fischer et al., 2018). The differences between colder glacial and warmer interglacial periods are driven by climate feedbacks alongside changes in the Earth's orbit which affect incoming radiation. The Last Interglacial or LIG, occurred 130,000-116,000 years ago. At 127,000 years ago, at high latitudes orbital forcing led to summertime top-of-atmosphere shortwave radiation 60–75 Wm$^{-2}$ greater than the PI period. Summer temperatures in the Arctic during the LIG are estimated to be around 4.5 K above those of today (CAPE members, 2006; Kaspar et al., 2005; IPCC, 2013; Capron et al., 2017). Prior to 2020, most climate models simulated summer LIG temperatures which were too cool compared with these LIG temperature observations (Otto-Bliesner et al., 2013; IPCC, 2013). This led Lunt et al. (2013); Otto-Bliesner et al. (2013) and IPCC (2013) to suggest that the representation of dynamic vegetation changes in the Arctic might be key to understanding LIG summertime Arctic warmth.

Guarino et al. (2020b) argued that loss of Arctic sea-ice in the summer could cause the warm summer Arctic temperatures, without the need for dynamic vegetation. Using the HadGEM3 model, which was the UK's contribution for the LIG CMIP6-PMIP4 project, Guarino et al. (2020b) found that the model simulated a fully sea ice-free Arctic during the summer, i.e. it had less than 1 mill. km$^2$ of sea ice extent at its minimum. This unique, near complete, loss of summer sea ice appears to happen in the UK model, because it includes a highly advanced representation of melt ponds (Guarino et al. 2020b; Diamond et al. 2021). These are shallow pools of water which form on the surface of Arctic





sea ice and which determine how much sunlight is absorbed or reflected by the ice (Guarino et al.,
2020b).

Malmierca-Vallet et al. (2018) found the signature of summertime Arctic sea ice loss in Greenland ice
cores. Kageyama et al. (2021) then led the international community in compiling all available marine
core Arctic sea ice proxy data for the LIG and testing it against CMIP6-PMIP4 simulations. The
Kageyama et al. (2021) synthesis of ocean core-based proxy records of LIG Arctic sea-ice change,
like Malmierca-Vallet et al. (2018), showed that compared to the PI it is very likely that Arctic sea ice
was reduced. However, Kageyama et al. (2021) also showed that directly determining sea-ice changes
from marine core data is difficult. The marine core observations suffer some conflicting
interpretations of proxy data sometimes from the same core, and imprecision in dating materials to the
LIG period in the high Arctic. Thus, determining the mechanisms and distribution of sea ice loss
during the LIG by directly inferring sea ice presence (or absence) from these preserved biological data
alone is not possible (Kageyama et al., 2021).

The Coupled Model Intercomparison Project Phase 6 (CMIP6) Paleoclimate Model Intercomparison
Project Phase (PMIP4) or CMIP6-PMIP4 LIG experimental protocol prescribes differences between
the LIG and PI in orbital parameters, as well as differences in trace greenhouse gas concentrations
(Otto-Bliesner et al., 2017). This standardised climate modelling protocol is therefore an ideal
opportunity for the community to use models to explore the causes of Arctic warmth using multi-
model approaches. In particular, it offers the opportunity to address the questions of whether the
Arctic sea ice loss is sufficient to explain LIG summertime temperature observations, or whether the
Arctic vegetation changes idea (Lunt et al., 2013; Otto-Bliesner et al., 2013; IPCC, 2013), is still
potentially required.

Guarino et al. (2020b) showed that the HadGEM3, the only CMIP-PMIP4 model with an ice-free
Arctic  at the LIG, has an excellent match with observed Arctic air temperature in summer. The
average ΔSSAT in HadGEM3, for all locations with proxy observations, is +4.9 ± 1.2 K compared



with the observational mean of +4.5 ± 1.7 K. This model also matched all, except one, marine core
sea-ice datapoints from Kageyama et al. (2021). Here we investigate whether there are more CMIP6-
PMIP4 models with a similarly good ΔSSAT and if so, whether other models with a good match also
suggest a much-reduced sea ice area (SIA) during the LIG. We further compute the correlation and
linear relationship in the models between ΔSSAT and ΔSIA and subsequently use this equation and
proxies for ΔSSAT to estimate ΔSIA. Section 2 describes the proxy data and models used in this
study as well as the analysis methods. The results are presented in Section 3 which first evaluates the
modelled PI and LIG sea ice distribution against observations and then use the above described
approaches to estimate the sea ice reduction at the LIG. Section 4 summarises the results and
discusses their shortcomings and implications.

**2. Data and methods**
**2.1 Observational data**
The LIG SSAT proxy observations used to assess LIG Arctic sea ice in the Guarino et al. (2020b)
study were previously published by CAPE members (2006); Kaspar et al. (2005) and 20 of them were
also used to assess CMIP5 models in the IPCC (2013) report. A detailed description of each
observation is available (CAPE members, 2006; Kaspar et al., 2005; IPCC, 2013; Capron et al.,
2017). Each observation is thought to be of summer LIG air temperature anomaly relative to present
day and is located in the circum-Arctic region; all sites are from north of 51°N. There are 7 terrestrial
based temperature records; 8 lacustrine records; 2 marine pollen-based records; and 3 ice core records
included in the original  IPCC (2013) compilation. Guarino et al. (2020b) added to this an additional
new observation from the NEEM Greenland ice core from Capron et al. (2017), bringing the total
number of proxies records to 21 (Table 1). Figure 1 shows the location, and type, for each numbered
observation. Whilst the exact timing of this peak warmth has not yet been definitively determined, it
is reasonable to assume that these measurements are approximately synchronous across the Arctic. It
is however very unlikely that the peak warmth was synchronous across both hemispheres (see Capron
et al. (2014); Govin et al. (2015)), and further investigation of the synchronicity of peak warmth





occurs across the Northern Hemisphere is merited. For consistency with modelled data, temperature
anomalies computed against present day conditions (i.e. 1961-1990 baseline) were corrected to
account for a +0.4K of global warming between PI (1850) and present day (1961-1990) conditions
(Turney and Jones, 2010). Therefore, Table 1 and Guarino et al. (2020b) values differ slightly (+0.4K)
from the original datasets so that they represent temperature anomalies relative to the PI.

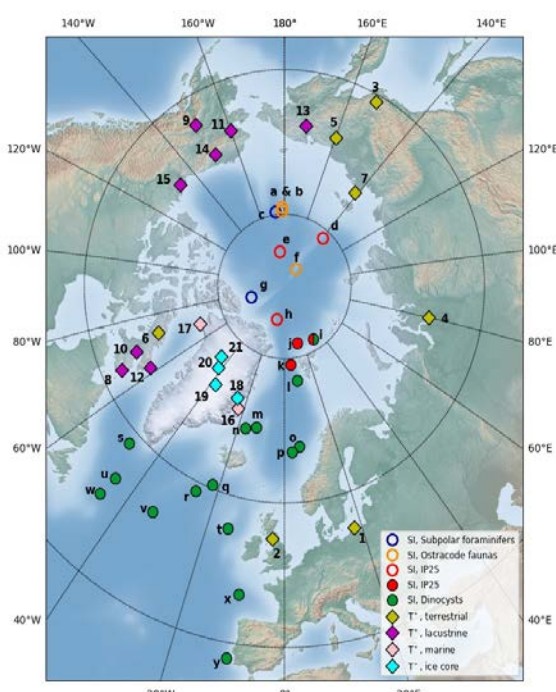


*Figure 1: Map of data locations numbered to match Table 1. This combines the Kageyama et al.*
*(2021) sea ice locations 1 to 20 alongside with the temperature proxies from Table 1.*

Most of the sites have temperature uncertainty (one standard deviation) estimates, which are provided
in the Table 1. However, for 9 sites, the standard deviation of the temperature data was not available.
A standard deviation of ± 0.5K was used to account for this missing uncertainty: this is the smallest
standard deviation found in any proxy record across all sites, and is thus as a conservative estimation
of the uncertainty associated to proxy data (Guarino et al., 2020b).




*Table 1: Compilation of LIG-PI summertime surface air temperature (SSAT) anomalies used by Guarino et al. (2020b).*

| Number | Lat | Lon | Site | Observation type | Observation (K) |
|---|---|---|---|---|---|
| 1 | 55 | 18 | Europe | Terrestrial: pollen, plant macrofossils | 3.4 ± 0.5 |
| 2 | 55 | -3 | UK | Terrestrial: Pollen, plant macrofossils | 2.4 ± 0.5 |
| 3 | 61 | 152.5 | Magadan | Terrestrial: pollen | 6.4 ± 2 |
| 4 | 68 | 80 | West-central Siberia | Terrestrial: pollen, plant macrofossils | 5.4 ± 2 |
| 5 | 68 | 160 | Northeast Siberia | Terrestrial: pollen | 6.4 ± 2 |
| 6 | 70 | -72.5 | Flitaway | Terrestrial: insects, plant remains | 4.9 ± 0.5 |
| 7 | 73.33 | 141.5 | Bolshoy Lyadhovshy | Terrestrial: pollen | 4.9 ± 0.5 |
| 8 | 63 | -66 | Robinson Lake | Lacustrine: pollen | 5.4 ± 0.5 |
| 9 | 64 | -150 | Birch Creek/ky11 | Lacustrine: pollen | 1.4 ± 1 |
| 10 | 66 | -69.2 | Amarok Lake | Lacustrine: pollen | 4.9 ± 0.5 |
| 11 | 67 | -160 | Squirrel Lake | Lacustrine: pollen, plant macrofossils | 1.9 ± 1.5 |
| 12 | 67 | -62 | Cumber | Lacustrine: pollen | 5.9 ± 1.5 |
| 13 | 67.5 | 172.08 | Lake Elgygytgyn | Lacustrine: pollen | 3.4 ± 1 |
| 14 | 69 | -151 | Ahaliorak Lake | Lacustrine: pollen | 1.9 ± 1.5 |
| 15 | 69 | -133 | Lake Tuk 5 | Lacustrine: plant macrofossils and beetles | 2.4 ± 0.5 |
| 16 | 71.75 | -23 | Jameson | Marine: pollen, plant macrofossils, beetles, other invertebrates | 5.4 ± 0.5 |
| 17 | 76.35 | -68.3 | Thule | Marine: pollen, chironomids | 4.4 ± 0.5 |
| 18 | 73 | -25 | Renland | Ice core: d18O, dD | 5.4 ± 0.5 |
| 19 | 73 | -38 | GISP2 | Ice core: d18O, dD | 5.4 ± 0.5 |
| 20 | 75 | -42 | NGRIP | Ice core: d18O, dD | 5.4 ± 0.5 |
| 21 | 76.4 | -44.8 | NEEM(ds) | Ice core: d18O, dD | 8 ± 4 |
| - | - | - | Arctic | Mean of observations 1 to 21 | 4.5 ± 1.7 |



## 2.2. Models and model output

We analyse Tier 1 LIG simulations, based on the standard CMIP6-PMIP4 LIG experimental protocol (Otto-Bliesner et al., 2017). The prescribed LIG (127 ka) protocol differs from the CMIP6 PI simulation protocol in astronomical parameters and the atmospheric trace GHG concentrations. LIG astronomical parameters are prescribed according to orbital constants (Berger and Loutre, 1991), and





atmospheric trace GHG concentrations are based on ice core measurements: 275 ppm for $CO_2$; 685
ppb for $CH_4$; and 255 ppb for $N_2O$ (Otto-Bliesner et al., 2017).

The CMIP6-PMIP4 model simulations were run following the Otto-Bliesner et al. (2017) protocol,
except CNRM-CM6-1, which used GHG at their PI values rather than using LIG values. For all
models, all other boundary conditions, including solar activity, ice sheets, aerosol emissions etc., are
identical to the PI simulation. In terms of the Greenland and Antarctica ice sheets, a PI configuration
for the LIG simulation is not unreasonable (Kageyama et al., 2021; Otto-Bliesner et al., 2020). LIG
simulations were initialized either  from a previous LIG run, or from the standard CMIP6 protocol PI
simulations, using constant 1850 GHGs, ozone, solar, tropospheric aerosol, stratospheric volcanic
aerosol and land use forcing. Whilst PI and LIG spin-ups vary between the models, with CNRM the
shortest at 100 years, most model groups aimed to allow the land and oceanic masses to attain
approximate steady state *i.e.* to reach atmospheric equilibrium and to achieve an upper-oceanic
equilibrium - which generally seems to take around 300 to 400 years. LIG production runs are all
between 100-200 years long, which is an appropriate length for Arctic sea ice analysis (Guarino et al.,
2020a).

Whilst fifteen models have run the CMIP6-PMIP4 LIG simulation (Kageyama et al., 2021; Otto-
Bliesner et al., 2020), and have uploaded model data to the Earth System Grid Federation (ESGF), we
exclude four simulations for the following reasons. The AWI-ESM and Nor-ESM models have LIG
simulations with two versions of model. To avoid undue biasing of results, we include only the
simulation from the latest version for each model. Additionally, for INM-CM4-8 model, no ocean or
sea ice fields were available for download, excluding this model  from our analysis. Finally, we
exclude the CNRM model in the analysis because apart from using PI instead of LIG GHG
concentrations and a short spin-up time, the model also has known issues with its sea-ice model. The
model produces much too thin sea ice in September and March compared with observational evidence
and the snow layer on the ice is considerably overestimated (Voldoire et al., 2019). As a possible
consequence of these issues, the CNRM model is also an outlier in an otherwise highly correlated





(inverse) relationship in the models between the LIG-PI albedo change over the Artic sea-ice and the
LIG-PI SSAT change over the ice, being the only model that produces a warmer LIG with almost no
reduction in albedo (Figure A1). While we consider the CNRM ice model unreliable for this study, we
note that the inclusion of the model in our analysis only reduces the correlation coefficients but does
not change the overall conclusions.

We thus analyse the difference between the PI and LIG simulations from eleven models. Out of the
eleven simulations of the LIG, seven have 200 years simulation length (data available to download in
ESGF), the remaining four are 100 years in length. For PI control runs, we use the last 200 years of PI
control run available in ESGF for each model. Details of each model: model denomination, physical
core components, horizontal and vertical grid specifications, details on prescribed vs interactive
boundary conditions, details of published model description, and LIG simulation length  (spin-up and
production runs) are contained in (Kageyama et al., 2021). Data was downloaded from the ESGF data
node: https://esgf-node.llnl.gov/projects/esgf-llnl/ (last downloaded on 23rd June 2021).

The spatial distribution of sea ice is usually computed in two ways, by its total area or its extent. The
sea ice extent (SIE) is the total area of the Arctic ocean where there is at least 15% ice concentration.
The total sea ice area (SIA) is the sum of the sea ice concentration times the area of a grid cell for all
cells that contain some sea ice. In this paper, the SIA refers to the SIA of the month of minimum sea
ice, as computed by using the climatology of the whole simulation.

**2.3. Assessing model skill to simulate reconstructions of ΔSSAT**
The model skill is quantified using two measures based on 1) the percentage of the 21 proxies for
ΔSSAT (in Table 1) for which the model produce a value within the error bars, and 2) the Root Mean
Square Error (RMSE) of the modelled SSAT compared to the proxies. To assess whether the model
match a proxy point, we compute summer mean (June to August) surface air temperatures for every
year for the PI and LIG for each model. Climatological summer temperature is the time mean of these





summer temperatures for the entire simulation length. Our calculated model uncertainties on the
climatological summer mean temperatures are one standard deviation of summer mean time series for
each model. Bilinear interpolation in latitude-longitude space was used to extract values at the
observation locations from the gridded model output. For climatological summer mean temperature, if
there is an overlap between observation SSAT (plus observational uncertainty) and the simulated
SSAT (plus model uncertainty) then, for that location, the result is considered as a match. Similarly,
the RMSE error is calculated using the modelled SSAT values averaged over the summer months of
the entire simulation length.

**3. Results**
**3.1. Simulated Arctic sea ice distribution**
The sea ice distribution in the models have been reported previously in Kageyama et al. (2021) and is
included here to make this work self-reliant. For the PI, the model mean value for summer minimum
monthly SIA is 6.4 mill. $km^2$. Due to a lack of direct observations for the PI, the PI model results are
compared with observed 1981 to 2002 satellite observations, keeping in mind that the modern
observations are for a climate with a higher atmospheric CO2 level of ~380 ppm, compared to the PI
atmospheric CO2 levels of 280 ppm. The modern observed mean minimum SIA is 5.7 mill $km^2$
(Reynolds et al., 2002). In general, the simulations show a realistic representation of the geographical
extent for the summer minimum. More models show a slightly smaller area compared to the present-
day observations, however EC-Earth, FGOALS-g3, and GISS170 E2-1-G simulate too much ice
(Figure 2). Overestimations appear to be due to too much sea ice being simulated in the Barents-Kara
area (FGOALS-g3, GISS-E2-1-G), in the Nordic Seas (EC-Earth, FGOALS-g3) and in Baffin Bay
(EC-Earth). Kageyama et al. (2021) also note that MIROC-ES2L performs rather poorly for the PI,
with insufficient ice close to the continents. The other models have a relatively close match to the
15% isoline in the NOAA Optimum Interpolation version 2 data (Reynolds et al., 2002; Kageyama et
al., 2021).





For the LIG, the model output is compared against the LIG sea ice synthesis of Kageyama et al.
(2021), which include marine cores collected in the Arctic Ocean, Nordic Seas and northern North
Atlantic (Figure 3). These data show that south of 79°N in the Atlantic and Nordic seas the LIG was
seasonally ice-free. These southern sea ice records provide quantitative estimates of sea surface
parameters based on dinoflagellate cysts (dinocysts). North of 79°N the sea-ice-related records are
more difficult to obtain and interpret. A core at 81.5°N brings evidence of summer being probably
seasonally ice-free during the LIG from two indicators: dinocysts and IP25/PIP25. However, an
anomalous core close by at the northernmost location of 81.9°N, with good chronology, shows IP25-
based evidence of substantial (> 75%) sea ice concentration all year round. Other northerly cores do
not currently have good enough chronological control to confidently date material of LIG age. All
models, except FGOALS, generally tend to match the results from proxies of summertime Arctic sea
ice in marine cores with good LIG chronology (Figure 3), apart from the anomalous northernmost
core for which the IP25 evidence suggest perennial sea ice (Kageyama et al., 2021). This may mean
that all the models tend to have similar problems in simulating Arctic sea ice during the LIG or that
the LIG IP25 signal in the Arctic indicates something else. What is clear is that a new approach with
other Arctic datasets, such as SSAT, may be needed to make progress on the LIG Arctic sea ice
question.










*Figure 2: Climatological Minimum PI sea ice concentration maps for each model. The first panel represents the multi model mean (MMM).*













*Figure 3: Climatological minimum LIG sea ice concentration maps for each model. Marine core results are from Kageyama et al. (2021): orange outlines indicate that the dating is uncertain; green outlines indicate the datapoint is from the LIG. The first panel represents the multi model mean.*



For the LIG, there is very little difference between the maximum (wintertime) Arctic SIA and that of
the PI (which is 15-16 mill. km$^2$ between the PI and the LIG in most models), but every model shows
a reduction in summer sea ice in the LIG compared to the PI (Table 2). Our model mean LIG
summertime Arctic is 2.9 mill. km$^2$, compared to 6.4 mill. km$^2$ for the PI, or a 55% PI to LIG
decrease. There is large inter-model variability for the LIG SIA during the summer (Figure 4). All
models show a larger sea-ice area seasonal amplitude for LIG than for PI, and the range of model SIA
is larger for LIG than for PI (Figure A2). The results for individual years show that no model is close
to the ice-free threshold for any model summer during their PI simulation (Figure 4) but for the LIG
summer SIA, there are three models which are lower than 1 mill. km$^2$ for at least one summer during
the LIG simulation (Figure 4). Of these three, HadGEM3, shows a LIG Arctic Ocean free of sea ice in
all summers, *i.e.* its maximum SIE is lower than 1 mill. km$^2$ in all LIG simulation years. CESM2 and
NESM3 show low climatological SIA values (slightly above 2 mill. km$^2$) in summer for the LIG
simulation, and both have at least one year with a SIE minimum which is below 1 mill. km$^2$, though
their average minimum SIE values are just below 3 mill. km$^2$. Of these low LIG sea ice models,
HadGEM3 and CESM2 realistically capture the PI Arctic sea ice seasonal cycle, whilst NESM3
overestimates winter ice and the amplitude of the seasonal cycle (Cao et al., 2018).

*Table 2: The minimum climatological sea ice area for the PI and the LIG, changes, and the*
*associated ΔSSAT anomalies. Percentage reductions are calculated from PI minimum SIA for each*
*model.*

| MODEL | SIA PI | SIA LIG | ΔSIA | SIA | ΔSSAT |
|---|---|---|---|---|---|
| (units) | (mill. km$^2$) | (mill. km$^2$) | (mill. km$^2$) | (% loss) | (K) |
| MMM | 6.36 | 2.93 | -3.43 | 53.87 | 3.6±1.3 |
| ACCESS-ESM1-5 | 5.48 | 2.39 | -3.09 | 56.44 | 2.6±1 |
| AWI-ESM-1-1-LR | 5.37 | 3.76 | -1.61 | 29.99 | 1.7±1.1 |
| CESM2 | 5.31 | 1.62 | -3.69 | 69.54 | 3.3±1 |
| EC-Earth3-LR | 8.86 | 3.65 | -5.21 | 58.84 | 5.7±2.6 |
| FGOALS-g3 | 8.83 | 5.55 | -3.29 | 37.19 | 4.8±1.5 |



| | | | | | |
|---|---|---|---|---|---|
| GISS-E2-1-G | 8.87 | 5.54 | -3.32 | 37.47 | 3.4±1.4 |
| HadGEM3-GC31-LL | 5.21 | 0.13 | -5.07 | 97.48 | 4.9±1.2 |
| IPSL-CM6A-LR | 6.42 | 2.46 | -3.96 | 61.74 | 4.4±1.2 |
| MIROC-ES2L | 4.20 | 2.79 | -1.41 | 33.66 | 2.1 ± 0.6 |
| NESM3 | 5.50 | 1.64 | -3.86 | 70.14 | 3 ±0.9 |
| NorESM2-LM | 5.92 | 2.75 | -3.17 | 53.52 | 3.6±1.1 |




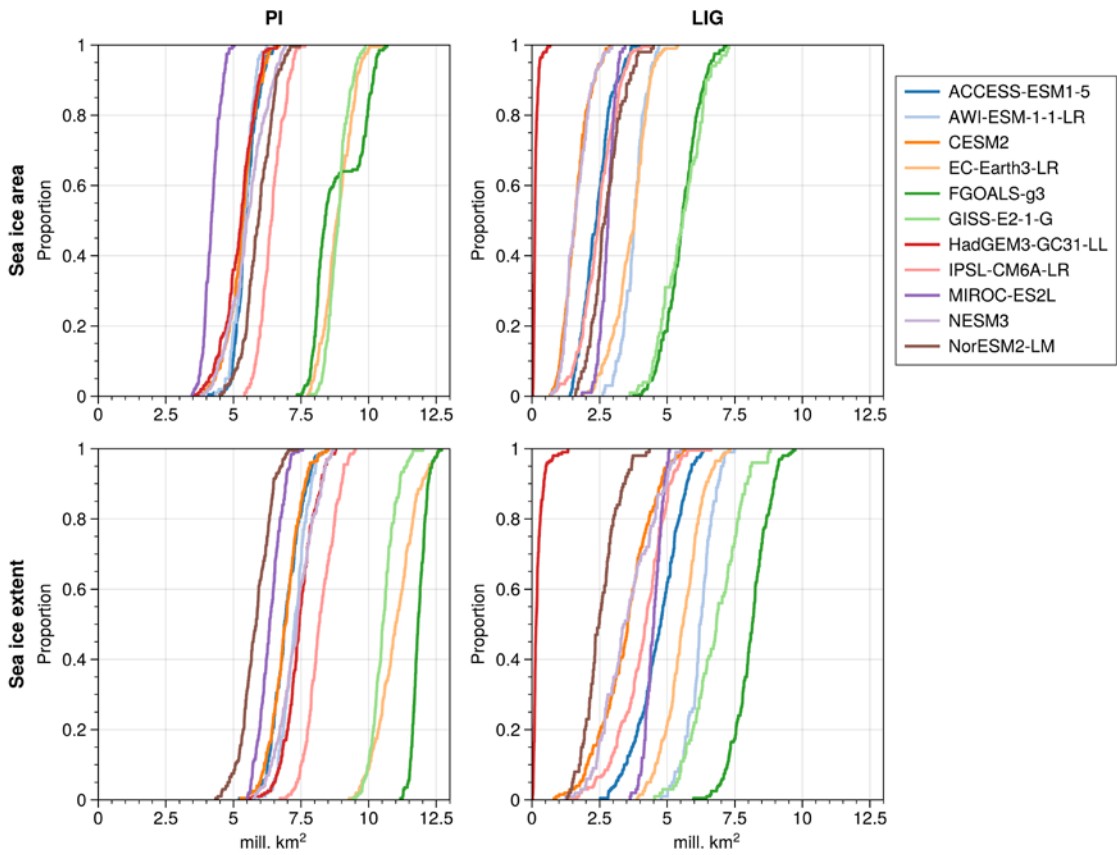

*Figure 4: Cumulative distribution of minimum SIA of individual years in LIG and PI simulations, i.e SIA versus proportion of years which fall below the corresponding SIA value. HadGEM3 has minimum SIA below 1 mill km² for all years in LIG runs. CESM2 has 6.5%, and NESM3 8%, LIG years with SIA below 1 mill km². Lower Panels are same but for SIE.*





## 3.2. Estimating ΔSIA from model skill to simulate ΔSSAT

We first investigate whether there is a relationship between how well models match proxy ΔSSAT and the magnitude of SIA reduction that they simulate for the LIG. A visual comparison of modelled ΔSSAT and proxy estimates for ΔSSAT is also shown in Figure 5. As described in Section 2, two different approaches are used to quantify the skill of the models to simulate ΔSSAT, based on 1) the RMSE of the model-data ΔSSAT at the proxy record locations and 2) the percentage ΔSSAT proxies that the model can correctly match, within model and data error. Here the focus is on quantifying model skill across all data records, but for reference, the model-versus-proxy ΔSSAT for each location is provided for each model individually in Figure A3. The RMSE skill estimate and the percentage match estimate provide very similar indications of which models have good skill to reproduce proxy ΔSSAT. The five models with the lowest RMSE also have the highest percentage match and the two models with the highest RMSE have the lowest percentage match (Figure 6). Both approaches show that the models with better skill to simulate ΔSSAT have a high absolute ΔSIA. The only outlier is EC-Earth, which has an average skill (6th best model of 11) but a high SIA reduction at the LIG. This occurs because the EC-Earth PI simulation has an excessive SIA, more than 3 million km$^2$ compared with observations; this enables it to have a large ΔSIA value, whilst likely retaining too much LIG SIA. Quantitively there is a correlation of r=-0.65 (p=0.03) between the magnitude of ΔSIA and the RMSE, and a correlation with r=0.67 (p=0.02) between the magnitude of ΔSIA and the percentage match of the model (Figure 6). Given that the SIA reduction from the PI to the LIG could be dependent on the starting SIA at the PI, we repeat the analysis for percentage SIA loss from the PI (rather than absolute SIA loss) and find that is correlates similarly to the model skill to reproduce ΔSSAT (Figure A4).







*Figure 5: Summertime surface air temperature (SSAT) anomaly (LIG - PI) maps for each model overlain by observed summer temperature anomalies. Proxies are detailed in Table 1 and Guarino et al. (2020b); colours are the same as used for the underlying model data. The first panel represents the multi model mean.*





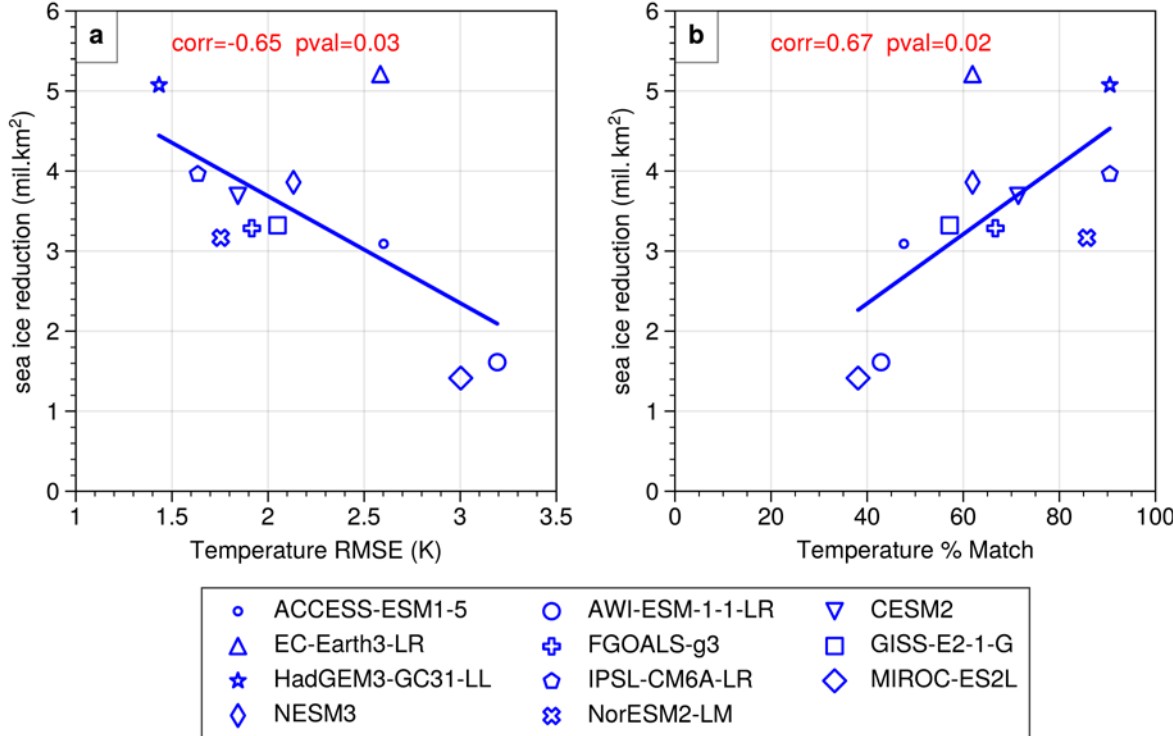

*Figure 6: Modelled magnitude of ΔSIA versus model skill to simulate proxy ΔSSAT. a) The modelled magnitude of ΔSIA is scattered against the RMS error of the modelled ΔSSAT compared to the proxy ΔSSAT for the 21 data locations. b) The modelled magnitude of ΔSIA scattered against the percentage of ΔSSAT data points that the model can match (see methods).*

In general, where models have a closer match with the ΔSSAT, they have a higher absolute ΔSIA, as well as a larger percentage reduction of SIA from the PI. We thus look at our best performing models for an indication of true LIG Arctic sea ice reduction. The four models with the best agreement of ΔSSAT to proxies are in order of skill; HadGEM3, IPSL, NORESM2, and CESM2. The top two performing models simulate an average SIA loss of 4.5 mill. km$^2$ from an average starting PI SIA of 5.8 mill. km$^2$ to a final LIG SIA of 1.3 mill. km$^2$, which equates to a percentage SIA loss of 79%. Including also the two next-best performing models in the average results in an average SIA loss of





4.0 mill. km$^2$ to a final LIG SIA of 1.7 mill. km$^2$ from an average starting PI SIA of 5.7 mill. km$^2$,
which equates to a percentage SIA loss of 71%.

The question arises as to why there is a linear relationship between model skill to simulate Arctic
ΔSSAT  and SIA reduction. One possibility is that the mean proxy ΔSSAT of 4.5 K is higher than
what most models produce, and that the warmer models are thus closer to the proxies and also more
likely to reduce sea ice. In the next section, this question is addressed by investigating whether ΔSIA
is closely related to ΔSSAT itself.

**3.3. Estimating ΔSIA from the modelled ΔSIA-ΔSSAT relationship and proxy ΔSSAT**
Here we investigate whether the models suggest a linear relationship between ΔSSAT and ΔSIA, and
if so, exploit that together with proxy ΔSSAT to estimate the most likely (true) value for ΔSIA. We
first calculate the mean ΔSSAT in the model at all 21 proxy data locations and compare it to the
magnitude of ΔSIA in each model (Figure 7a). The two are well correlated with r=0.86 (p=0.001) and
the regression equation provide a dependence of ΔSIA on ΔSSAT. Using this relation, the observed
mean ΔSSAT at the proxy locations points to a SIA reduction of 4.4 mill. km$^2$ from the PI. This
constitutes a 77% reduction from the present day observation of 5.7 mill. km$^2$, which is also the
average SIA for the PI in the two most skilful models identified in the previous section. Using this
value for the PI sea ice, suggests remaining minimum of 1.3 mill. km$^2$  of sea ice during the LIG
summer. An average LIG minimum of 1.3 mill. km$^2$  implies that some LIG summers must have been
ice-free (below 1 mill. km$^2$ in SIE) but that most summers would have had a small amount of sea ice.




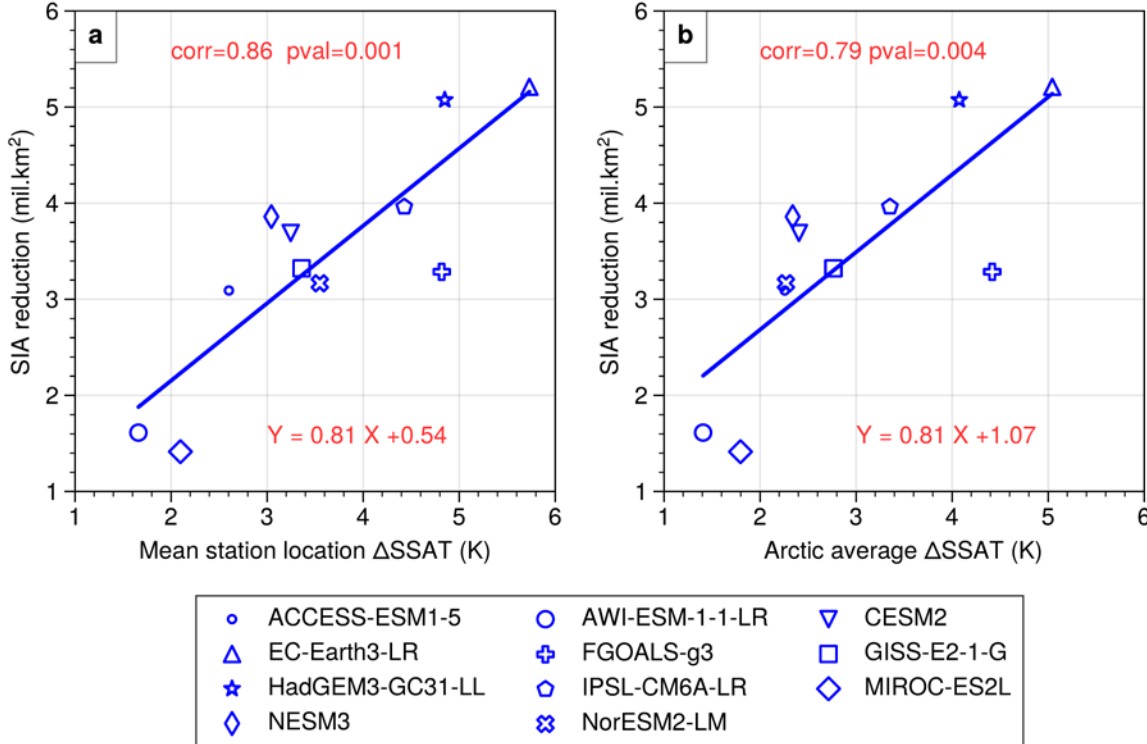

*Figure 7: Modelled magnitude of ΔSIA versus modelled ΔSSAT for the Arctic. a) The modelled ΔSIA is scattered against mean modelled ΔSSAT at the 21 data locations. b) The modelled ΔSIA is scattered against the mean modelled ΔSSAT averaged over the Arctic north of 60°N.*

The ΔSSAT relationship to ΔSIA has so far been computed using the mean ΔSSAT at the locations of the data. To test whether this method would also work for the Arctic in general, the ΔSSAT is next averaged over the whole Arctic north of 60°N and compared with ΔSIA (Figure 7b). The correlation between ΔSSAT and ΔSIA is a somewhat reduced when calculating ΔSSAT across the whole Arctic, though it is still highly significant (r=0.79, p=0.004). An estimate for proxy-based Arctic-wide ΔSSAT can be derived by applying the close relationship between Arctic ΔSSAT and station ΔSSAT in the models (Figure 8, r=0.97, p <0.001). Inserting the ΔSSAT averaged over all proxy-records, of 4.5 K, in the regression equation in Figure 8, gives an estimate for proxy-based Arctic-wide ΔSSAT





of 3.7±0.1 K. Applying the regression equation in Figure 7b and using this estimate for Arctic-wide
ΔSSAT suggests a PI to LIG sea ice reduction of 4.5 mill. km$^2$, which is very similar to the estimate
derived from the station data alone (of 4.4 mill. km$^2$).

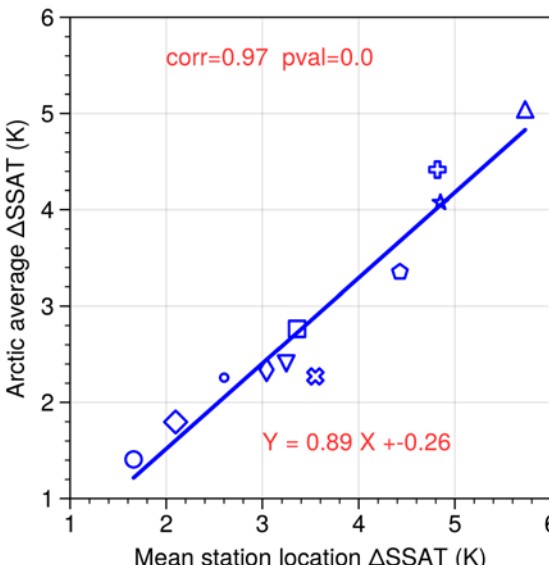

Figure 8: Modelled Arctic-wide ΔSSAT versus modelled mean ΔSSAT at the data locations for the 11
models.

**4. Discussion and conclusions**
As discussed in the introduction, neither proxies nor modelling results alone allow currently for a
convincing estimate of the Arctic sea ice reduction at the LIG. Here we apply a joint approach to
make progress. We deduce how much sea ice was reduced during the LIG, using 11 of the most recent
CMIP6-PMIP4 LIG model simulations and proxy observations of summer air temperature changes.
The reduction of sea ice from the PI to the LIG in the models range from 30% to 96% with an average
of 55%. No model is close to the ice-free threshold, of maximum SIE lower than 1 mill. km$^2$, for any
model year-summer during their PI simulation. During the LIG, the HadGEM3 model is the only one
that has an Arctic Ocean free of sea ice in all summers, although CESM2 and NESM3 show SIA



values of around 2 mill. km$^2$, in association with intermittently ice-free conditions. We found that
larger LIG SIA reduction from the PI is related to greater SSAT warming, the two being correlated
with r=0.86 across the models. In particular, the 8 models with largest SIA reduction are all able to
match, within uncertainty, the mean PI to LIG summertime Arctic warming of 4.5 ± 1.7 K at the 21
proxy locations. This magnitude of warming was difficult to reach with previous generations of LIG
models.

We find that the good match between the (ice-free) HadGEM3 and the Guarino et al. (2020b) summer
Arctic temperature dataset is not unique. However, we find that it is not random either and that there
is a correlation between model skill to match the ΔSSAT and the reduction of SIA from the PI to the
LIG (both when using an RMSE skill test and when using a best-match skill test). The two most
skilful models simulate an average LIG sea ice area of 1.3 mill. km$^2$ which is a 4.5 mill. km$^2$ or 79%
reduction from their PI values.  Whilst we cannot assume all model error ΔSSAT is attributable to
ΔSIA, it is reasonable to assume that the better performing models for ΔSSAT are also better at
simulating ΔSIA, because of the close relationship between warming and sea ice loss.

Some of the proxies are more difficult for the models to simulate (Figure 9 and Figure A3). In
particular, it appears that the Greenland ice core SSAT value from NEEM of +8 K (observation 21 in
Table 1 Figure 9) is higher than any model simulates; though with a ±4 K uncertainty it is
nevertheless matched by some models. Terrestrial proxies three and six, with SSAT values of +6.4 K
are also only rarely matched. Further work on the observational side would be useful. These LIG
SSAT proxy reconstructions were used in the IPCC (2013) report and by Guarino et al. (2020b); and
were previously published by IPCC (2013); CAPE members (2006); Kaspar et al. (2005); Capron et
al. (2017). Thus, this dataset should ideally be improved. One start point for this would be adding
uncertainties to the (nine) sites which do not currently have these numbers.



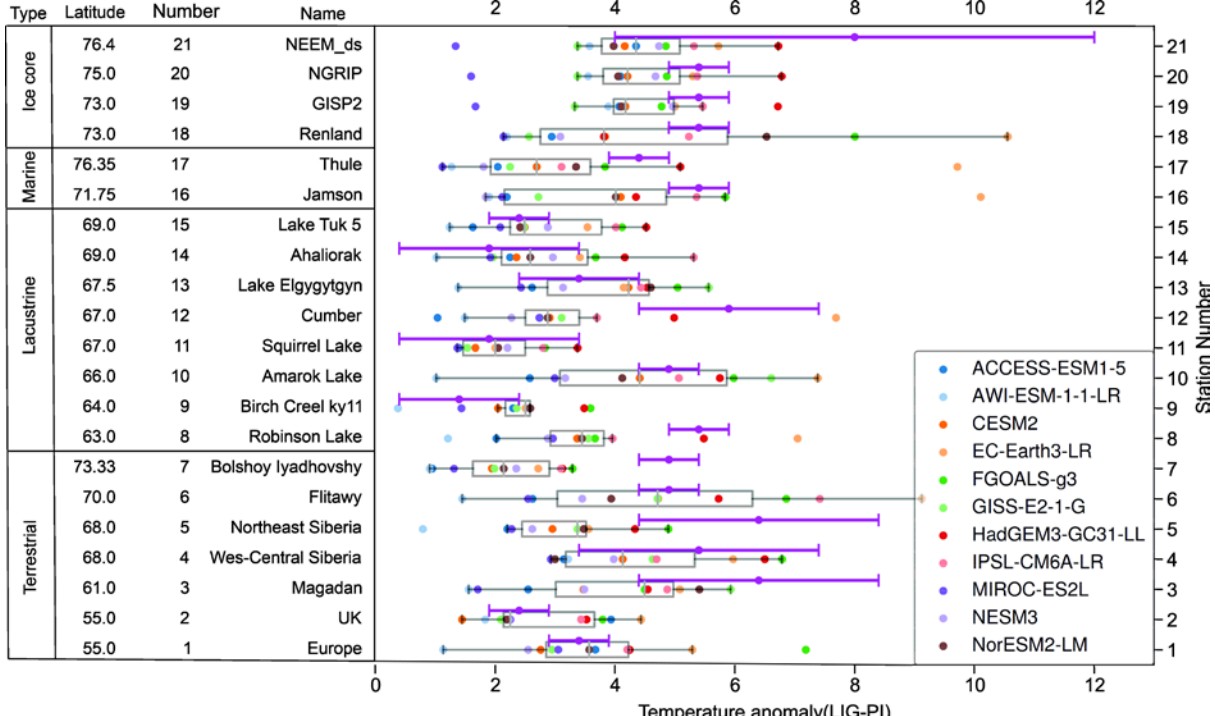


*Figure 9: Proxy ΔSSAT (violet dots and uncertainty bars) and simulated ΔSSAT for all models (coloured dots) for each proxy record location (rows). Grey boxes extend from the 25th to the 75th percentile of each locations distribution of simulated values and the vertical lines represent the median.*


The correlation between model skill to simulate ΔSSAT and the magnitude of ΔSIA is convincing (r=
0.66 and p= 0.003 on average for the two skill tests). However, the two quantities are not
straightforward to relate through a dynamical process. On the other hand, it is well known that there is
a positive feedback between Arctic temperature and Arctic sea-ice, with warmer temperatures more
likely to melt sea ice, and less sea ice producing a smaller albedo to incoming solar radiation and so
less cooling from solar reflection. This dynamic is evident in the strong correlation of r=0.86 between
the magnitude of ΔSIA and ΔSSAT. The reconstructed ΔSSAT from proxies, of 4.5 ±1.7 K, is larger
than most models simulate, so the models that match the ΔSSAT most closely would be the models
with a larger ΔSSAT than average and thus also a larger ΔSIA. The only model that has a large SIA





reduction and not a good skill to match SSAT is EC-Earth, which features a PI simulation with far too
much sea ice, which allows an excessive LIG to PI Arctic warming. An additional result of our study
is that the mean ΔSSAT at the proxy locations is strongly correlated to Arctic-wide ΔSSAT north of
60°N in the models (r=0.97). Applying the regression relation between the two, implies that the mean
ΔSSAT at the proxy locations, of  4.5 K, is equivalent to an Arctic-wide warming at the LIG of 3.7 K.
This is thus a more representative value for the Arctic warming at the LIG, than using the simpler
proxy-location average.

The strong linear correlation between the magnitude of ΔSIA and ΔSSAT is applied to the proxy-
reconstructed ΔSSAT to give an estimate of the reduction of SIA from the PI to LIG of 4.4 mill. km$^2$,
similar to that derived from our "best skill" approach. A similar value of 4.5 mill. km$^2$ is obtained
when extrapolating the method to Arctic-wide ΔSSAT north of 60°N. The models and data have
uncertainties, and the regressions applied are not between perfectly correlated quantities. However, it
is clear from both applied methods (each with two variants) that proxy-reconstructed ΔSSAT, in
combination with the model output, implies a larger sea ice reduction than the climatological multi-
model mean of 55%. It suggests a LIG SIA of ~1.3 mill. km$^2$, which is consistent with intermittently
ice-free summers – but with (low ice area) ice-present summers likely exceeding the number of ice-
free years.  This result suggests that the fully-ice free HadGEM3 is somewhat too sensitive, and loses
summer sea ice too readily during the LIG, alongside that most other PMIP4 models are insufficiently
insensitive do not lose enough sea ice.

*Code availability.* Python code used to produce the manuscript plots is available on request from the
authors.

*Data availability.* The summer air temperature dataset is available at https://data.bas.ac.uk/full-
record.php?id=GB/NERC/BAS/PDC/01593. All model data is available from the ESGF data node:
https://esgf-node.llnl.gov/projects/esgf-llnl/.





**Appendix**

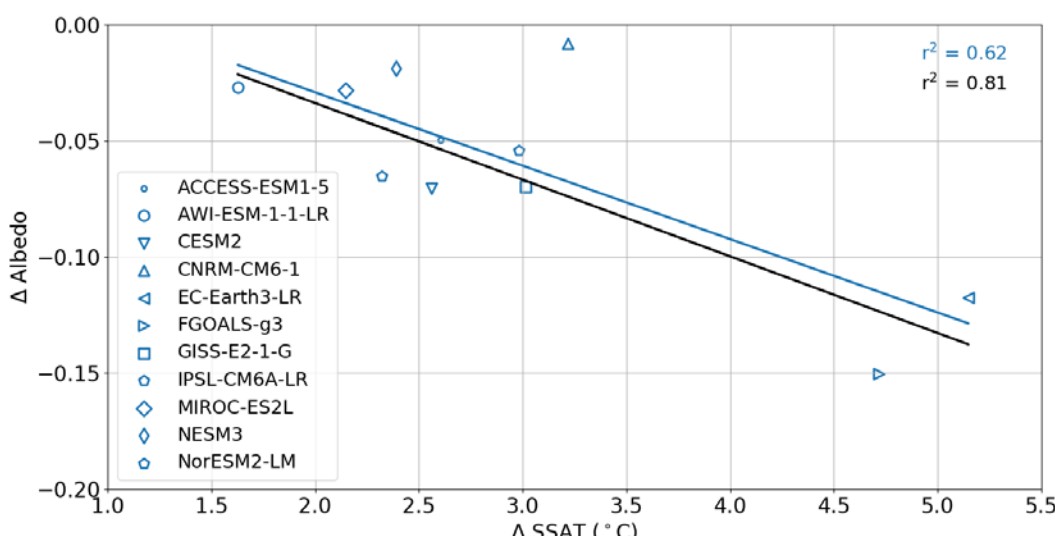



Figure A1. LIG-PI change in albedo over Arctic sea-ice as a function of LIG-PI change in SSAT (°C)
over the ice. The $r^2$ values and the linear fit lines are for the models including CNRM (blue) and
excluding CNRM (black). The CNRM model (upside triangle) is an outlier that influences the
strength rather than the nature of the correlation.





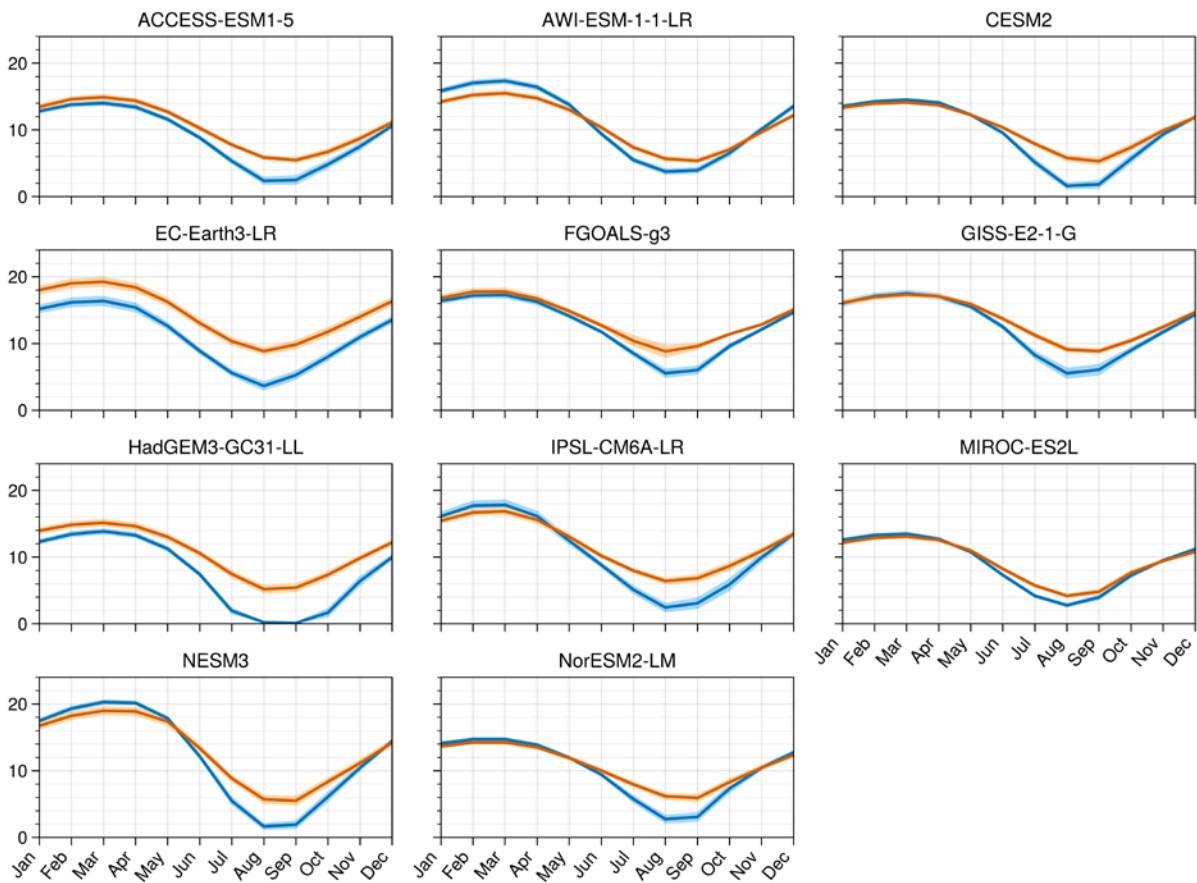


Figure A2. Sea ice area climatological seasonal cycle for each model.



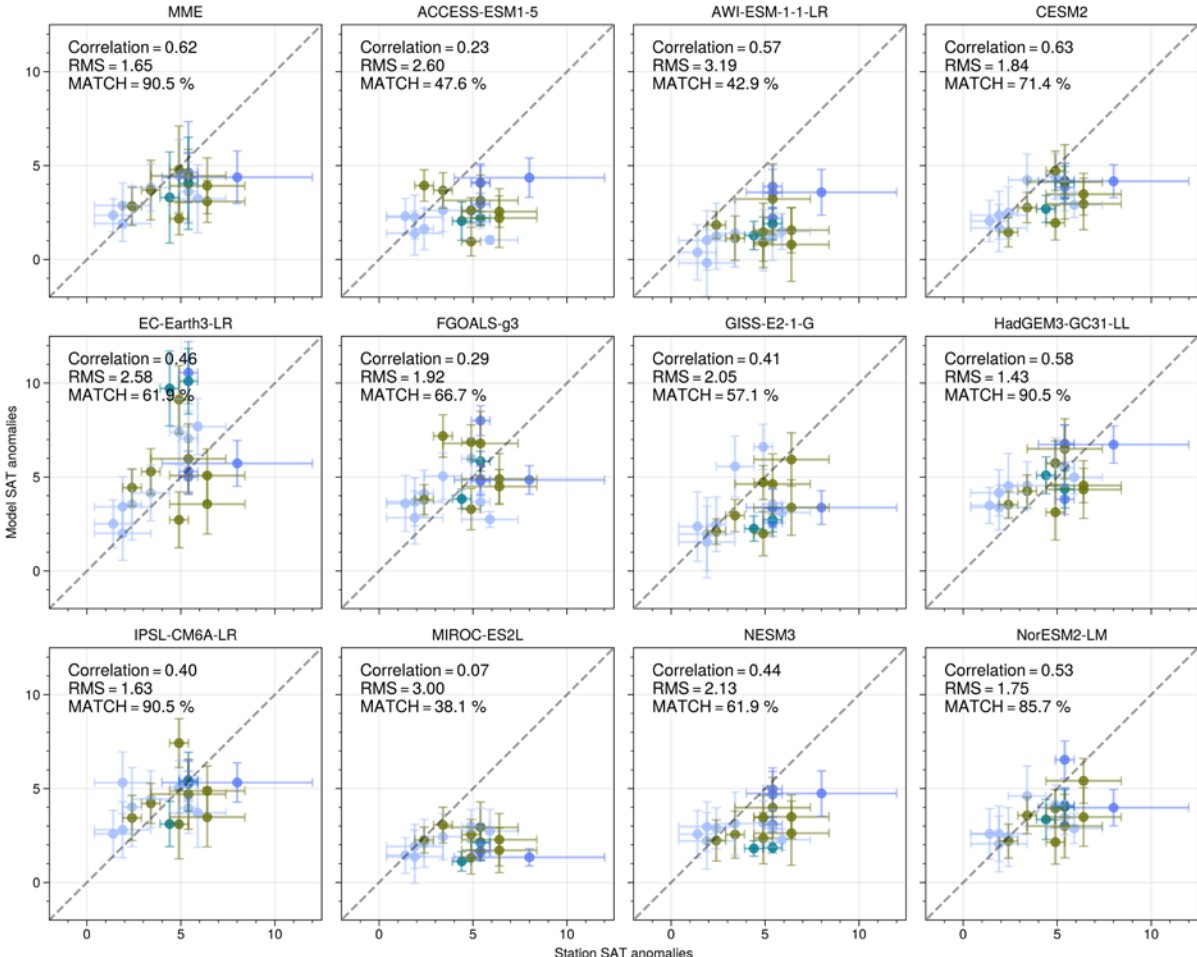

Figure A3. Modelled ΔSSAT versus proxy ΔSSAT. The scatter points show model data versus observations for each proxy location. Error-bars represent one standard deviation on either side of the proxy estimate. The correlation coefficients, between X and Y, RMSE and percentage matches with observations for each model are indicated in each panel.



470

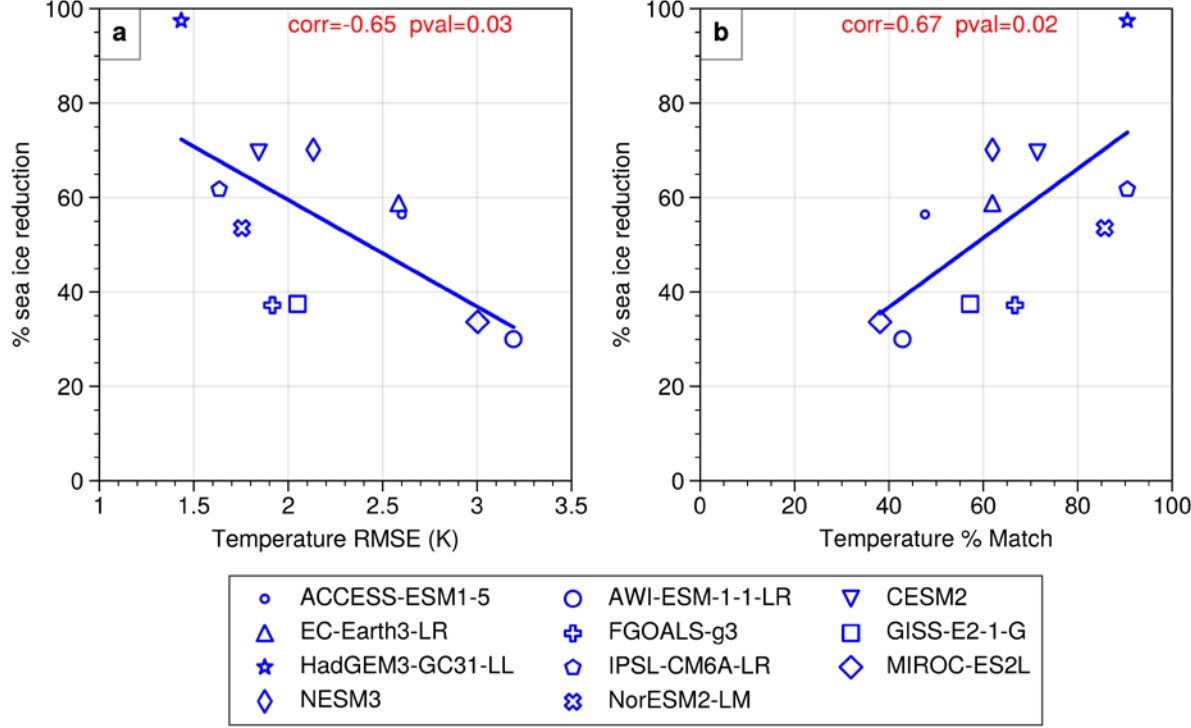

471

Figure A4: Modelled % sea ice area reduction from the LIG to the PI versus model skill to simulate proxy ΔSSAT. a) The modelled %SIA reduction is scattered against the RMSE of the modelled ΔSSAT compared to the proxy ΔSSAT for the 21 data locations. b) The modelled % SIA reduction scattered against the percentage of ΔSSAT data points that the model can match (see methods).






Figure A5. Scatter Plot for climatological ΔSSAT at each observational location versus climatological
ΔSSAT averaged over entire Northern Hemisphere in each model



*Author contributions*. LCS planned and wrote the original draft. RS analysed model results and
prepared the figures. Figure 1 which was prepared by IVM.  AdB wrote the second draft. MS
undertook additional analysis, checks and researched particular model results. All authors contributed
to the final text.

*Competing interests*. The authors have no competing interests.

*Acknowledgements*. LCS and RS acknowledge the financial support of NERC research grant
NE/P013279/1 and NE/P009271/1. LCS and IVM have received funding from the European Union's
Horizon 2020 research and innovation programme under grant agreement No 820970. AdB and MS
were supported by Swedish Research Council grant 2020-04791. This work used the ARCHER UK
National Supercomputing Service (http://www.archer.ac.uk) and the JASMIN analysis platform
(https://www.ceda.ac.uk/services/jasmin/).



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
