# Peer review of "Summer surface air temperature proxies point to near sea-ice-free conditions in the Arctic at"

_EGUsphere, 2022_

## Referee Comment (RC1)

Review of manuscript by Sime et al. entitled 'Summer surfce air temperature proxies point to near sea-ice-free conditions in the Arctic at 127ka'.

The authors present a well though-out and well written study to study LIG sea-ice changes and simulated relationships between changes in temperature and sea ice. I have only a few comments which are listed below.

**Main comment:**
To make this work more directly relevant to a larger public and to the understanding of ongoing climate change, it would be usefull to compare the presented findings with observational records. In particular, I'm wondering if from observational estimates one can also deduce a relationship between changes in Arctic summer temperatures and sea-ice area, and how such a relationship would compare to the LIG-based estimate. A dSIA of 4.4 mil. km2 for a dSSAT of 3.7, gives dSIA of 1.18 mil. km2 per 1K dSSAT. Potentially, there is a difference because the LIG climate was close to equilibrium while an observational estimate would be based on transient climate change. Another problem could be that the LIG winter forcing was negative, while the present-day GHG-driven winter forcing is positive. Nonetheless, this could be a nice addition to the presented work.

**Minor comments:**
Line 25: So dSIA corresponds to NH-wide changes in sea ice while dSSAT corresponds to local, proxy location changes in temperature? Reword to make this clear.

Lines 85-88: You make it sound like it is either sea-ice changes or vegetation changes, while in reality they will very likely go hand-in-hand. It is hard to imagine a sea-ice free Arctic that is several degrees warmer in summer, that does not see any changes in vegetation on the surrounding continents. And indeed, these vegetation changes will likely feedback on the temperature and sea-ice changes.

Figure 2: why not show the observational sea-ice concentrations on all the maps, or in a separate plot?

Lines 299-302: by using percentage changes, is EC-Earth no longer an outlier as mentioned above, or is it still an outlier?

Figure A4: using this metric is seems HadGEM3-GC31-ll is now an outlier!

**Technical comments:**
Line 91: maybe better to use 'reconstructed' instead of 'observed' throughout the text to avoid confusion with recent and present-day observations? Or alternatively, consequently use 'proxy observation' as you do on for instance line 106?

Figure 1: the figure and legend shows empty and filled red circles, but they appear both to be SI, IP25. What is the difference between them? Why are the sea-ice reconstructions shown on this map, are they used in this study?

Line 267: remove 'for any model'?

Lines 285-287: perhaps reverse the order of 1 and 2 to be coherent with earlier mentioning.

Figure 8: add legend to the figure or mention in the caption that the legend can be found in figure 7.

Figure A5: indidate what is on the x-axis and y-axis.

---

## Author Comment (AC1)

Review of manuscript by Sime et al. entitled 'Summer surface air temperature proxies point to near sea-ice-free conditions in the Arctic at 127ka'.

**The authors present a well though-out and well written study to study LIG sea-ice changes and simulated relationships between changes in temperature and sea ice. I have only a few comments which are listed below.**

**Main comment:**

**To make this work more directly relevant to a larger public and to the understanding of ongoing climate change, it would be useful to compare the presented findings with observational records. In particular, I'm wondering if from observational estimates one can also deduce a relationship between changes in Arctic summer temperatures and sea-ice area, and how such a relationship would compare to the LIG-based estimate. A dSIA of 4.4 mil. km2 for a dSSAT of 3.7, gives dSIA of 1.18 mil. km2 per 1K dSSAT. Potentially, there is a difference because the LIG climate was close to equilibrium while an observational estimate would be based on transient climate change. Another problem could be that the LIG winter forcing was negative, while the present-day GHG-driven winter forcing is positive. Nonetheless, this could be a nice addition to the presented work.**

We thank #1 for their kind review of our manuscript.

In answer to their main comment, we agree that comparing our presented findings with observational records is indeed a useful exercise. In accordance with the practice of most recent papers looking at present-day sea ice in the Arctic, we use NSIDC sea ice data and ERA5 SAT data from the satellite-era (1979-2020). The figure below shows the scatter plot of these SAT versus SIA data. The calculated relationship gives an (observational) dSIA of 1.32 mil. Km2 per 1K. This relationship shows slightly higher values for dSIA than LIG one of 1.18 mil. km2 per 1K dSSAT. In accordance with the comment, we will add this comparative present-day observation relationship figure to the manuscript in the supplementary section, and provide the comparative numbers in the main manuscript.

Whilst we agree with the #1 that potentially, there could be difference because the LIG climate was (probably) close to equilibrium while the observational estimate is based on transient climate change, the close agreement between the LIG and present-day relationship tend to support the idea that Arctic sea ice and SAT tend to be both reside in the fast reacting part of the climate, and that slower ocean and vegetation (transient) feedbacks therefore  seem to play a lesser role is determining the association between the two.

[Figure]

*Figure R1:* Scatter plot of SAT versus SIA for current period. JJA surface air temperature versus NH September Sea ice area for each year from 1979-2020.  Anomalies computed from year 1979 values. SIA is from NSIDC (https://nsidc.org/data/g02135/versions/3) and Air temperature (area averaged north of 60ºN) is from ERA5 reanalysis (Hersbach et al. 2020).

**Minor comments:**

**Line 25: So dSIA corresponds to NH-wide changes in sea ice while dSSAT corresponds to local, proxy location changes in temperature? Reword to make this clear.**

We propose that in line 25 (in abstract) "Second, across the 11 models, the averaged ΔSSAT at the 21 proxy locations is inversely correlated with ΔSIA (r = -0.86)"  be changed to "Second, across the 11 models, the averaged ΔSSAT at the 21 proxy locations as well  the pan Arctic average delta SSAT, is inversely correlated with ΔSIA (r = -0.86 and 0.79 respectively).

**Lines 85-88: You make it sound like it is either sea-ice changes or vegetation changes, while in reality they will very likely go hand-in-hand. It is hard to imagine a sea-ice free Arctic that is several degrees warmer in summer, that does not see any changes in vegetation on the surrounding continents. And indeed, these vegetation changes will likely feedback on the temperature and sea-ice changes.**

**1 is correct that in reality sea-ice and vegetation changes will go hand in hand. However we do think it is useful to point out that sea-ice change on its own is sufficient to explain the LIG SSAT changes. We thus propose to modify the text so that "In particular, it offers the opportunity to address the questions of whether the Arctic sea ice loss is sufficient to explain LIG summertime temperature observations, or whether the Arctic vegetation changes idea (Lunt et al., 2013; Otto-Bliesner et al., 2013; IPCC, 2013), is still potentially required." becomes instead, "In particular, the existing non-dynamic-vegetation PMIP4 LIG protocol and associated simulations offer the opportunity to address the question of whether the Arctic sea ice loss alone is sufficient to explain LIG summertime temperature observations, or whether active vegetation modelling, and the idea of vegetation feedbacks (Lunt et al., 2013; Otto-Bliesner et al., 2013; IPCC, 2013) are required. This said, we recognize that in reality there must also be LIG Arctic vegetation feedbacks. These should be explored in future modelling work."**

**Figure 2: why not show the observational sea-ice concentrations on all the maps, or in a separate plot?**

The reason we do not do this is that we would be comparing modelled pre-industrial SIA with observed historical SIA data. This is rather confusing, so we prefer not to do this.

**Lines 299-302: by using percentage changes, is EC-Earth no longer an outlier as mentioned above, or is it still an outlier?**

**1 is correct to point out that using percentage changes EC-Earth is not an outlier (Figure A4).**

**Figure A4: using this metric is seems HadGEM3-GC31-ll is now an outlier!**

Given that HadGEM3 has approximately 100 % summertime sea ice loss during the LIG, it is indeed an outlier: no other model simulates such a complete loss of sea ice. That said, if the amount of sea ice loss and observational agreement is considered HadGEM3 fits ok, being the model with the best LIG SSAT match – but with the largest sea ice loss. This is also pointed out in Kageyama et al (2021) and Diamond et al (2021).

**Technical comments:**

**Line 91: maybe better to use 'reconstructed' instead of 'observed' throughout the text to avoid**

Will do.

**confusion with recent and present-day observations? Or alternatively, consequently use 'proxy observation' as you do on for instance line 106?**

Will do.

**Figure 1: the figure and legend shows empty and filled red circles, but they appear both to be SI, IP25. What is the difference between them?**

On the F1 open/filled symbols, open symbols correspond to records with uncertain chronology, and filled symbols correspond to records with good chronology, following the Kageyama et al (2021) convention. Apologies that this was not clarified in the caption. We will add this information during revision.

**Why are the sea-ice reconstructions shown on this map, are they used in this study?**

**1 is correct, these sea ice reconstruction data are shown in Figure 3.**

**Line 267: remove 'for any model'?**

Will do.

**Lines 285-287: perhaps reverse the order of 1 and 2 to be coherent with earlier mentioning.**

Thank you to #1 for pointing out this. Since the figures follow the order mentioned here, we will reverse the order mentioned in text in lines 194-196 to be consistent with this.

**Figure 8: add legend to the figure or mention in the caption that the legend can be found in figure 7.**

We will add to the caption that the legend can be found under Figure 7.

**Figure A5: indicate what is on the x-axis and y-axis.**

Thanks to #1 for also spotting this. We will incorporate this change in the revised version.

**Additional references:**

Hersbach, H, Bell, B, Berrisford, P, et al. The ERA5 global reanalysis. Q J R Meteorol Soc. 2020; 146: 1999– 2049. https://doi.org/10.1002/qj.3803

---

## Author Comment (AC2)

Comment on egusphere-2022-594

**Anonymous Referee #2**

Referee comment on "Summer surface air temperature proxies point to near sea-ice-free conditions in the Arctic at 127 ka" by Louise Claire Sime et al., EGUsphere, https://doi.org/10.5194/egusphere-2022-594-RC2, 2022

**This manuscript discusses the Arctic sea ice conditions at 127 ka based on the relationship between sea ice and temperature, it is an interesting topic and well written. However, there are still some questions need to be further discussed.**

We thank #2 for their kind comments and helpful review.

**1. Due to the sea surface temperature (SST) is more related to sea ice than the surface air temperature (SAT), why the SAT is chosen instead of SST?**

Much like sea ice, there are very few SST proxy records from the Arctic from the LIG period. Kageyama et al. (2021) discusses the reasons for the lack of Arctic Sea surface records from that time. In summary though, it is largely due to difficulties with dating Arctic marine cores. For this reason, Guarino et al. (2020) and this manuscript focus on using what is available for the LIG, which are SSAT proxy records. That is why we compare these available SSAT proxy reconstructions of LIG with model simulations, effectively extending the work of Guarino et al, and investigating relationship between SIA and the surface temperatures across all the PMIP4-LIG simulations. By taking this multi-model approach, we can obtain more robust conclusions about sea ice and Arctic climate during LIG.

**2. A short summary about why these proxy records are considered to represent the summer surface air temperature should be given in order to better understand the model- data comparison.**

We thank #2 for the suggestion. Some more detail about the dataset is given in Guarino et al. (2020). We will add some extra explanation in the revised manuscript, drawn largely from the original CAPE (2006) synthesis paper:

"Terrestrial climate can be reconstructed from diagnostic assemblages of biotic proxies preserved in lacustrine, peat, alluvial, and marine archives and isotopic changes preserved in ice cores and marine and lacustrine carbonates (CAPE, 2006; Guarino et al., 2020). Quantitative reconstructions of climatic departures from the present-day are derived from range extensions of individual taxa, mutual climatic range estimations based on groups of taxa, and analogue techniques (CAPE, 2006). These proxy records are considered to represent the summer surface air temperature because summer temperature is also the most effective predictor for most biological processes, though seasonality and moisture availability may influence phenomena such as evergreen vs. deciduous biotic dominance (Kaplan et al., 2003)."

**3. In lines 210-217, if the simulations show a realistic representation of the geographical extent for the summer minimum, the CO2 increases 100 (280 to 380) ppmv. The summer minimum SIA decreases 0.7 (6.4 to 5.7) mill. km2. How do you think about the sensitivity of Arctic sea ice in response to CO2?**

In this study, where we are comparing LIG and Pre-Industrial simulations, CO2 concentrations are not very different (prescribed in models as 276 and 280 ppm respectively). Hence the changes are not likely from CO2 forcing.

In Kageyama *et al.* (2021) Section 4.3 discuss in more depth the relationship between response to LIG climate forcings and transient CO2 forced responses in models by comparing LIG results with transient 1pco2 experiments (Figure 12 in their paper). They found that the models that respond strongly to LIG forcing also respond strongly for the 1pctCO2 forcing, and the model with the smallest response for the LIG has the smallest response to the 1pctCO2 forcing. For #2's interest, Notz et al. (2016) also shows in observed sea ice (present day) has a very linear relationship with CO2.

We will add these points to the revised manuscript in the discussions section.

**4. In part 3.1, different models show significant difference in the simulated Arctic sea ice for both the PI and LIG simulations. What do you think leads these difference between different models? How about the sensitivity of Arctic sea ice in response to astronomical forcing and how about the polar amplification in different models due to both of them have a great effect on the Arctic sea ice?**

Sea ice formation and melting can be affected by a large number of factors inherent to the atmosphere and the ocean dynamics, alongside the representation of sea ice itself within the model (*i.e.* the type of sea ice scheme used). In coupled models it can therefore be difficult to identify the causes of this coupled model behavior (Kagayama et al. 2021, Sicard et al,2022). Nevertheless Kagayama et al. (2021; Section 4), alongside Diamond et al. (2021) do address the question of what drives model differences in summertime LIG sea ice. In summary:

- 1. All models show a major loss of summertime Arctic sea ice between the PI and LIG.
- 2. Across all models, there is an increased downward short-wave flux in spring due to the imposed insolation forcing and a decreased upward short-wave flux in summer, related to the decrease of the albedo due to the smaller sea ice cover. Differences between the model results are due to a difference in phasing of the downward and upward shortwave radiation anomalies.
- 3. The sea ice albedo feedback is most effective in HadGEM3. It is also the only model in which the anomalies in downward and upward shortwave radiation are exactly in phase.
- 4. The CESM2 and HadGEM3 models (which both simulate significant sea ice loss) exhibit an Atlantic Meridional Overturning Circulation (AMOC) that is almost unchanged between PI and LIG, while in the IPSLCM6 model (with moderate sea ice loss) the AMOC weakens. This implies that a reduced northward oceanic heat transport could reduce sea ice loss in the Central Arctic in some models.
- 5. The two models (HadGEM3 and CESM2) which had the lowest sea ice loss contain explicit melt pond schemes, which impact the albedo feedback in these models. Diamond et al. (2021) show that that the summer ice melt in HadGEM3 is predominantly driven by thermodynamic processes and those thermodynamic processes are significantly impacted by melt ponds.

On polar amplification, Fig R2 (below) shows the relationship between Arctic Amplification index and SIA changes. It is evident from the figure that the models have diverse response in Arctic amplification and a linear relationship between Arctic amplification index and sea ice change amongst models is not very evident.

**Figure R2**: Arctic Amplification (AA) index plotted against  $\Delta$ SIA (a) and percentage reduction of LIG sea ice relative to PI (b). AA index is defined as the ratio of the  $\Delta$ SSAT averaged over Arctic (north of 60°N) to that averaged over whole Northern hemisphere.

We will add these points as a separate section as Intermodel differences in the supplementary material.

5. Although your results show that near sea-ice-free conditions in the Arctic at 127 ka, some records indicate that there still exists substantial sea ice (for example, in lines 233-234 and in Stein et al. (2017) https://doi.org/10.1038/s41467-017-00552-1). More discussion about these records should be given.

Indeed #2 is correct that some marine core records suggest that there were perennial sea ice above Arctic core sites. The most up-to-date synthesis and discussion on marine core records is given in Kagayama et al. (2021). We show this same synthesis in our Figures 3.

Quoting from Kagayama et al. (2021): "Based on IP25/PIP25 records obtained from central Arctic Ocean sediment cores (see Fig. 1 for core locations and Table 1 for data), perennial sea ice cover probably existed during the LIG in the Central Arctic, whereas along the Barents Sea continental margin, influenced by the inflow of warm Atlantic Water, sea ice was significantly reduced (Stein et al., 2017). However, Stein et al. (2017) emphasizes that the PIP25 records obtained from the central Arctic Ocean cores indicating a perennial sea ice cover have to be interpreted cautiously as the biomarker concentrations are very low to absent (see Belt, 2018 for further discussion). The productivity of algal material (ice and open water) must have been quite low, so that (almost) nothing reached the seafloor or is preserved in the sediments, and there must have been periods during the LIG when some openwater conditions occurred, since subpolar foraminifers and coccoliths were found in core PS51/038 and PS2200 (Stein et al., 2017). It is however unclear whether these periods equate to more than 1 month yr -1 of open water (or seasonal ice conditions). This explains why some sites show both seasonal and perennial interpretations at the same site."

We do not suggest repeating all in the current manuscript. However, we suggest in response to #2 that we add a sentence after L 236 "models generally tend to match the results from proxies of summertime Arctic sea ice in marine cores with good LIG chronology (Figure 3), apart from the anomalous northernmost core for which the IP25 evidence suggest perennial sea ice (Kageyama et al., 2021). Stein

et al. (2017) suggest that PIP25 records obtained from the central Arctic Ocean cores indicating a perennial sea ice cover have to be interpreted cautiously, given that biomarker concentrations are very low to absent, so it is difficult to know how much weight to place on this particular result. Additionally, given Hillaire-Marcel et al. (2017) question the age model of the data from the central Arctic Ocean, thus these IP25 data need to be interpreted with some caution."

6. In lines 389-391, you state that " the 8 models with largest SIA reduction are all able to match, within uncertainty, the mean PI to LIG summertime Arctic warming of 4.5 ± 1.7 K at the 21 proxy locations". But in lines 397-399, "The two most skillful models simulate an average LIG sea ice area of 1.3 mill. km2 which is a 4.5 mill. km2 or 79% reduction from their PI values", only the average result of the two models is given, why not the average result of these eight models?

Thanks for pointing out the possible confusion which may arise while reading. To clarify further, the first sentence (lines 389-391) be rewritten as

"In particular, 8 out of 11 models are able to match, within uncertainty, the average PI to LIG summertime Arctic warming of 4.5 ± 1.7 K as recorded by surface temperature proxies. Among the models, two of them capture the magnitude of the observed dSSAT in more than 60% of the total proxy locations. These models simulate an average LIG sea ice area of 1.3 mill. km2 which is a 4.5 mill. km2 or 79% reduction from their PI values."

**7. It is not clear that how many model results are used to establish the relationship between $\Delta$ SSAT and $\Delta$ SIA, 2 or 8 or 11? If it is 2 or 11, why not 8?**

We used all the 11 models used in this study to derive the relationship between SSAT and SIA. (The discussion with 8 models will be corrected in response to the previous question, which will clarify the confusion raised in this question)

**8. The forcing mechanism for the near sea-ice-free conditions in the Arctic at 127 ka should be discussed**

Guarino et al (2020), Kageyama et al (2021), and Diamond et al (2021) discusses in detail about mechanisms for reduced sea ice in LIG simulations. Please see also our response to question 4.

**Additional references:**

Sicard, M., A.M. de Boer, and L.C. Sime 2022, Last Interglacial Arctic sea ice as simulated by the latest generation of climate models, Past Global Changes Magazine, 30(2): 92-93

D. Notz, J. Stroeve, Observed Arctic sea-ice loss directly follows anthropogenic CO2 emission. Science 354, 747–750 (2016).

Belt, S.: Source-specific biomarkers as proxies for Arctic and Antarctic sea ice, Org. Geochem., 125, 277–298, https://doi.org/10.1016/j.orggeochem.2018.10.002, 2018.

---

## Author Response (AR2)

Minor comments:

Considering that both the simulated relationships and the reconstructed temperatures have uncertainties, I think these should be more clearly reflected in a full uncertainty on the combined model-data estimate of LIG sea-ice area changes. For instance, the authors report on an average PI to LIG summertime Arctic warming of 4.5 ± 1.7. There is thus a substantial uncertainty in this estimate. Using the model-based relationships that are described (plus uncertainty), what is the resulting uncertainty in the sea-ice area estimates when taking into account this 1.7K uncertainty?

**We have now added the uncertainty values for the sea ice estimates computed from regression equations in the manuscript.**

**We also realised that a typo in our initial sea ice estimate resulting in an offset of 0.2 km$^2$ in the sea ice values from regression equations. This has been corrected in the manuscript.**

Lines 443-444: repeat the model-based LIG relationship value here to help the reader directly compare the observational-based estimate.

**Added in to sentence starting from line 448 as**
**This dynamic relationship is evident in the models for LIG, where there is a strong correlation of r=0.86 between the magnitude of ΔSIA and ΔSSAT amongst the models, and the intermodel relationship suggests sea ice decrease of 1.9 mill km$^2$ per 1K temperature rise (from the regression equation in Figure 7b)**

Technical comments:

Line 311: estimatns (line 320 in revised with corrections) **'estimations'**
Line 421: Ko (Line 432) **Corrected the sentence**